# Citizens’ Perception of COVID-19 Passport Usefulness: A Cross Sectional Study

**DOI:** 10.3390/bs12050140

**Published:** 2022-05-12

**Authors:** Jorge de Andrés-Sánchez, Mario Arias-Oliva, Jorge Pelegrin-Borondo

**Affiliations:** 1Department of Business Management, Social & Business Research Laboratory, Universitat Rovira i Virgili, 43002 Tarragona, Spain; mario.arias@ucm.es; 2Management and Marketing Department, Complutense University of Madrid, 28040 Madrid, Spain; 3Economics and Business Department, University of La Rioja, 26006 Logroño, Spain; jorge.pelegrin@unirioja.es

**Keywords:** COVID-19, immunity passport, utility perception of immunity passports, COVID-19 restrictions, quantile regression

## Abstract

This paper assesses the influence on people’s perception of the utility of the immunity passport (IP) program by sociodemographic factors, infectivity status, and the objective of its use. The material of this paper is a cross-sectional survey of 400 residents in Spain. The relation between utility perception and input variables is fitted with ordinary least squares (OLS) regression and linear quantile regression (LQR). The principal explanatory variable of usefulness perception is being vaccinated, especially when the objective of the IP is regulating mobility. The OLS estimate of the coefficient regression is (cr) = 0.415 (*p* = 0.001). We also found a positive and significant influence of that factor in all LQRs (cr = 0.652, *p* = 0.0026 at level (τ) = 0.75; cr = 0.482, *p* = 0.0047 at τ = 0.5 and cr = 0.201, *p* = 0.0385 at τ = 0.25). When the objective of the IP is regulating leisure, being vaccinated is relevant only to explain the central measures of usefulness perception. If the IP is used to regulate traveling, variables related to interviewees’ infectivity have greater relevance than sociodemographic factors. When its objective is ruling assembly, the more important variables than being vaccinated are gender and age. To create an effective implementation of the IP, it is advisable to have a general agreement among the population on its convenience. Therefore, the findings in this study have important implications for public health decision-makers.

## 1. Introduction

Until controlling COVID-19 was possible by using effective drugs and vaccines, measures to limit mobility and assembly were the only health-efficient action to mitigate its transmission [1]. Thus, since the rise of the SARS-CoV-2 pandemic, practically all countries have used extreme solutions, such as lockdowns and people’s domiciliary isolation [2,3]. Unfortunately, these measures pose a social burden from an economical point of view [4,5,6]. 

The existence of effective vaccines allows for immunizing a significant proportion of the population. Likewise, reliable COVID-19 tests prove that a person is not infected, albeit temporally, despite not being immune or that he/she is immune due to the presence of COVID-19 antibodies in the blood (with serological tests). Therefore, immunity passports (IPs) allow for documenting the infectivity status of an individual [7]. This certificate can prove that an individual is immune to SARS-CoV-2, or at least is not infected on a set date. Further, this could be because the person who has been vaccinated against COVID-19 and has recovered from the disease or has received a negative test result [8]. Different instances have advocated for the relaxation of restrictions limiting COVID-19 to IP owners since they have a low capability to spread the disease [4]. This relaxation can be applied in two contexts: traveling, and accessing public places and social activities [9]. The use of such a document would allow a better balance between reducing disease transmissibility and the undesirable collateral consequences of constraints in freedom of movement and freedom of assembly. In this regard, the International Air Transport Association (IATA) outlined the need for the IP to document passengers’ immunity status regarding SARS-CoV-2 [10]. Since July 2021, EU countries have issued a coordinated so-called COVID-19 certificate. This certificate collects whether the person has been vaccinated against SARS-CoV-2 and/or has recovered from COVID-19; circumstances that can be considered states of “permanent” immunity. It also certifies the diagnostic tests that have been carried out and their result in such a way that a negative result would prove a situation of temporary sterility. The IP document can be obtained on paper or in a digital version and can be stored on a mobile device [11].

The implementation of IPs raises several operational, ethical, and moral challenges that we cannot avoid [12]. In fact, empirical work by Aranzales et al. [13] outlines that a fair consensus is not agreed upon in the scientific community about the potential health and economic benefits, and societal costs of the ethical issues of an immunity certification program. Some issues that we can highlight are as follows:Several authors think that the IP must only be considered as a cause of permanent immunization after being vaccinated. In this case, the IP will become a vaccination certificate [7,14]. On the other hand, some opinions are favorable to considering immunity for those persons who have recovered from COVID-19 [15]. In this regard, it must be outlined that in many countries, e.g., such as in Spain, people who were infected by SARS-CoV-2 must wait several weeks after their recovery to receive a vaccine shot [16].The immunization duration of the COVID-19 vaccines is under study. However, it is commonly accepted that vaccine protection declines throughout time [17].The information management and certificate issuance systems must be reliable and robust in such a way that they cannot endanger data privacy or facilitate tampering with the certificates [7,9].Once the immunological requirements are met, IPs should be easy to obtain. In this regard, we must outline that if it is only possible to obtain certificates digitally through smartphones, older people who do not have the habits and skills to use these devices could experience a great barrier to obtaining an IP [14].The IP could be used in the labor market to discriminate against those workers without permanent immunity. Therefore, in countries where the health system is private, social groups with smaller incomes could be discriminated against because they cannot access vaccination [7,9].Those people who, due to physiological reasons, cannot receive the vaccine could also be discriminated against. This same reflection can be extended to citizens from the poorest countries that will not have acceptable vaccination coverage for several years. The time needed to reach an acceptable vaccination rate in all countries is a critical issue. Despite all adults having been vaccinated at the end of 2021 in developed countries [18], this fact will not follow in Africa until 2023.The mandatory use of an IP to allow social activities could be in conflict with rights linked to privacy [7,9]. Vaccination and COVID-19 test results could be classified under personal health data, and its use is prohibited by the ninth article of the General Data Protection Regulation (GDPR). The exception could be related to the public interest, such as controlling the COVID-19 outbreak [19].The mandatory use of an IP can also violate other personal rights, such as freedom of movement, especially for people who would not be able to take the vaccine because of health constraints or cannot access it [20]. Other rights that are in conflict with the implementation of an IP are religious and ideological freedoms [7,9].

Another issue to take into account is the relevance to the user’s utility perception of the context in which an IP must be adopted. In this regard, it must be differentiated in its use to limit COVID-19 transmission via mobility and because activities done in public places have empirical relevance [13,20]. 

The questions raised in the above paragraphs motivate the present empirical study. We evaluated the perception of the need to implement the IP, examining sociodemographic and contextual factors in regards to COVID-19 that influence perception, through the use of a survey of 400 residents in Spain. The Scopus and WoS databases show a great deal of literature evaluating citizens’ perceptions of the usefulness of different measures against COVID-19, such as the use of a mask and hand hygiene [21,22,23,24,25,26,27], the limitation of assembly and movement freedom [28], tracing apps [29,30,31,32] or vaccination [33,34,35]. However, the number of empirical studies regarding the acceptance of an IP and its implications is scarcer [20,36,37,38].

Concretely, our research seeks to answer the following research questions:

Research question 1 (RQ1): *Are people’s utility perceptions of the IP program utility different depending on the use (regulating travels* vs. *controlling access to public places and social activities) that is going to be given to that document?*

Research question 2 (RQ2): *What are the factors (gender, age, educational level, and those related to contextual situations linked with immunity to COVID-19) relevant to explaining utility perception toward the use of the IP?*

## 2. Materials and Methods

### 2.1. Materials

This paper analyzes a survey of residents of Spain over 18 years of age in Madrid and its close urban geographical area. The interview period was from 16 April 2021 to 29 April 2021. We have to outline that Spain, at the beginning of the spring of 2021, began a deep public debate about the COVID-19 passport that still continues. 

Trained interviewers, located throughout social networks or by phone, contacted people to interview. They supplied the surveyed persons with a Google Form link. Only a response per electronic device was allowed. The survey was answered without the interviewer’s supervision, and it did not demand any data that could identify interviewees, for example, their names or passport numbers. Therefore, the anonymity of the collected data was ensured. All participants were people of legal age. Voluntary completion of the questionnaire was taken as consent for the data to be used in research, and informed consent of the participants was implied through the survey’s completion. As we carried out the survey, it was not possible to determine whether a contacted person who received the link to answer actually responded to the questions. The study has received the approval of the Ethical Committee of Rovira i Virgili University (CEIPSA-2021-PR-0042).

In that survey, we established gender quotas (50% men and 50% women) and age quotas (18–24 years = 25%; 25–44 years = 25%; 45–64 years = 25%; plus 64 years = 25%). Therefore, despite all complete responses being 452, after considering the gender and age quotes, we used the first 400 surveys that attained those constraints. Notice that this paper presents a similar age distribution to other studies that embrace age rank 18+, such as [29,37]. The median age was 44.5 years and the average = 45.38 years (standard deviation = 19.87 years). These parameters are a consequence of the age quotes and present similar values to those in the survey of the Spanish population used in the international assessment of the IP’s implementation [38]. In this last paper, the average age in the Spanish sample was 48 years, and the standard deviation = 16. 

In Table 1, we can see that 59.5% of people have a university degree, 21.25% of respondents were vaccinated, 22.5% possessed natural immunization and 67.5% reported getting tested at least once. Notice, however, that this high rate of the population with a university degree is above that in the EU, which is slightly above 50%. The reason may be that the survey was done in an urban area with a high density of population and university institutions. Therefore, the not-so-great standard deviation of age could be linked to the fact that the respondents came from a located geographical area and have relatively homogenous high-level studies. The low vaccination rate in our sample is due to the survey being carried out in April 2021 and with that date, in Spain, only people above 60 years and professionals with a great deal of public exposure, such as health workers, had the opportunity to get vaccinated. More in-depth information about sociodemographic characteristics and immunity situations of surveyed people is summarized in Table 1.

### 2.2. Variables

The exposition in the introduction allows for stating that the evaluation of the utility of an IP program is a construct that embeds several dimensions. Some examples are as follows: its effectiveness to control the transmission, the easiness of its use for all age groups (e.g., often older people struggle with new tech), or possible conflicts with the right of privacy in personal information. Accordingly, we measured the response variable, the perception of the usefulness (USEFUL), by adapting the four questions from the scale in [39], which was applied to evaluate new technologies. These questions were formulated as “*I feel that implementing IP as a prophylactic measure is (1) a bad/good idea; (2) a foolish/wise idea; (3) an ineffective/effective measure; (4) a negative/positive measure*”. Notice, however, that the answers to these questions are not necessarily the same or similar. For example, an individual may give a high score to “ineffective/effective measure” because he/she perceives that IP implementation can allow good control over COVID-19 from spreading; however, they may also provide a low score for the item “foolish/wise idea” because his/her perception is that IP implementation damages individual rights.

The items were evaluated on an eleven-point scale that ranks from 0, the worst opinion, to 10, the best. Therefore, 5 implies a neutral position. These questions were responded to in two different contexts: the IP is used to regulate traveling; the IP is used to rule leisure activities. Table 2 shows descriptive statistics of these items, differentiating travel and assembly contexts.

To explain utility perception, we used three usual sociodemographic variables for this kind of study: gender (GENDER), age (AGE), and cultural status (CS), which have been widely tested to explain the perceptions and attitudes toward prophylactic measures against COVID-19 [21,22,23,24,25,26,28,29,30,31,34], about vaccination [33,34,35] and more scarce, in regard to the IP [37,38]. Contextual situations in regard to COVID-19 are also relevant to explaining COVID-passport perception [38]. Therefore, we also considered three variables linked to the personal situation, with regard to SARS-CoV-2 infection: If the individual is vaccinated (VAC); if he/she has a natural immunity because of past infection by SARS-CoV-2 (NATIM), and finally; the habit of obtaining diagnostic tests for COVID-19. This last variable is measured by the number of PCRs (NPCR) until the date of the survey. These variables are modeled as follows:Gender (GENDER) = a dichotomous variable that takes a value of 0 for males and 1 for females.Age (AGE) = a dummy variable that takes a value of 0 if the interviewee is under 45 years and 1 otherwise. In our sample, this is the median of that variable (see Table 1). Likewise, following [40], from that age, approximately, the probability that SARS-CoV-2 causes severe disease and death increases exponentially.Cultural status (CS) = a dummy variable whose value is 0 if the surveyed person has not completed a university degree and 1 if he/she is, at least, a graduate.Vaccination status (VAC) = a dichotomous variable that takes 0 if the person has not been vaccinated and 1 otherwise.Natural Immunization (NATIM) = a dichotomous variable that takes 0 if the person was never infected by SARS-CoV-2 and 1 otherwise.Number of PCRs done (NPCR) = takes 0 if the person never did any test and 1 otherwise.

### 2.3. Methods

To answer RQ1, we ran two statistical tests on every question about utility perception: a Student’s *t*-test on the means difference and a Wilcoxon signed-rank test (WSRT). Note, however, that we worked with two pairwise samples since the same person made a response to the hypothesis that the IP is used to regulate mobility and leisure activities in public places. This evaluation is expressed on a scale that ranges from 0 to 10. 

To assess RQ2, in a first step, we checked the internal consistency of the scales by using the usual measures: Cronbach’s alpha (CA), average variance extracted (AVE), and factor analysis. In the second step, the variable UTILITY was quantified by aggregating the value of its four items (see Table 2) by taking their standardized factor loading. This is a reliable way when factor analysis is applied to these items satisfactorily [41]. In the third and fourth steps, we fitted a linear regression on utility perception (UTILITY), over GENDER, AGE, CS, VAC, NATIM and NPCR, which were defined previously. Therefore, in these two steps, the regression model to be estimated is:
UTILITY = a_0_ + a_1_ × GENDER + a_2_ × AGE + a_3_ × CS + a_4_ × VAC + a_5_ × NATIM + a_6_ × NPCR
(1)

where coefficients a_1_, a_2_, …, a_6_ measure the influence of each explanatory variable on UTILITY. In the third step, (1) is adjusted by means of ordinary least squares (OLS). The last (fourth) step consists of fitting again (1) by means of linear quantile regression (LQR), whose use in social sciences was proposed in the seminal work by Koenker and Basset [42]. LQR offers robust statistical estimates that OLS does not make when the hypotheses of normality and homoscedasticity of errors are violated. Moreover, quantile regression is less sensitive to outliers [42]. Likewise, LQR allows describing not only how input factors impact explained variables around central distribution values but also in extreme responses. Therefore, LQR provides a more complete picture of the relations between variables missed in the methods, such as least squares [43]. In our paper, we fit LQRs at the probability levels of τ = 0.25, 0.50, and 0.75. Therefore, OLS and LQR, at the 0.5 level, estimate the impact of input factor variables around a central measure of UTILITY (the mean in the case of OLS and the median in the LQR at τ = 0.50). On the other hand, whereas LQR at τ = 0.75 may allow for assessing the impact of factors in the responses showing an IP acceptance above central mainstream, the results from LQR at τ = 0.25 allow assessment of the influence of explanatory variables in the perceptions that display a greater resistance towards the mandatory use of an IP than the central tendency.

## 3. Results

Table 2 shows that all items of the response variable present a mean and median > 5, i.e., it seems that the average evaluation of the use of an IP in our sample is closer to being positive than negative. We can also check that the IP has a better evaluation when it is used to mitigate COVID-19 spreading due to mobility than when it regulates access to public activities. However, this fact only has statistical significance in the third question, ineffective/effective (*p* = 0.009 for the Student’s t-test on mean differences and *p* = 0.019 in the case of WPRT).

The results in Table 3 suggest an internal consistency for the scale on UTILITY, since the factor loadings are >0.7, CA > 0.7, and AVE > 0.5. Notice that AVE < 0.9 and the factor loadings are high but relatively far from 1, i.e., despite the set of items measuring utility being highly correlated, they do not show redundancy. 

Table 4 shows the results of the OLS regression in the two contexts where an IP could be used. In the case of mobility regulation, the results of the overall fit are R^2^ = 0.0376 with Snedecor’s F = 2.56 (*p* = 0.0229). The regression adjusted for public events attains a weaker significance level (R^2^ = 0.0299, Snedecor’s F = 2.02 and *p* = 0.062). In both cases, the residuals are far from normal since the null hypothesis (i.e., normality) is rejected (*p* < 0.001). Likewise, in the regression linked to traveling, homoscedasticity of errors is also fairly rejected (*p* = 0.0229). However, this fact is less clear in the adjustment linked with the use of the IP to constrain access to public activities (*p* = 0.0643).

In Table 4, we can see that in both the OLS adjustments, being vaccinated is a key variable in explaining UTILITY. When the IP regulates traveling, the regression coefficient (rc) of VAC is 0.415 (*p* = 0.001) and the 95% confidence interval (95%CI) is [0.165, 0.665]. In the case of leisure activities, rc = 0.27 (*p* = 0.033) and 95%CI = [0.023, 0.517]. Likewise, being female (GENDER) also has a significant positive relation with UTILITY if the objective of the IP is regulating leisure activities (rc = 0.21, *p* = 0.038, 95%CI = [0.013, 0.407]).

Table 5 and Table 6 show the results of quantile regressions. To state goodness of fit in an LQR setting, it is not advisable to use a conventional R^2^, since it is a measure built for least square regression [44], and goodness of fit links predicted the conditional expectation with the observed values. Therefore, we measured the goodness of fit with pseudo-R^2^ by Koenker and Machado [44]. We have checked that in LQRs, the measure shows similar values to the conventional R^2^ in OLS regressions. Regarding the use of an IP to control the spread of COVID-19 in travel, Table 5 shows the following:LQR in median (τ = 0.5) attains similar results to OLS. The principal explanatory variable is VAC, which showed a positive significant impact on UTILITY (cr = 0.482, *p* = 0.0047, 95%CI = [0.150, 0.815]). However, having immunity due to a previous infection also had a negative significant impact on the median value of UTILITY (cr = −0.327, *p* = 0.0446, 95%CI = [−0.645, −0.009]).Being vaccinated also had a significant impact on UTILITY at τ = 0.25 (*p* = 0.0026, 95%CI = [0.230, 1.074]) and τ = 0.75 (*p* = 0.0385, 95%CI = [0.011, 0.391]). Therefore, it seems that VAC is the key variable to explain UTILITY when IP is used to limit COVID-19 transmission due to mobility.NPCR and cultural status have a positive significant impact on the lower and upper quantiles of UTILITY, respectively. Therefore, whereas in the quantile τ = 0.25 we obtain for NPCR a cr = 0.484 (*p* = 0.0162, 95%CI= [0.091, 0.877]) in LQR at τ = 0.75, CS attains a cr = 0.201 (*p* = 0.0106, 95%CI = [0.048, 0.355]).If IP limits assembly in public spaces, Table 6 reports the following:Despite still being relevant in this second context, being vaccinated is less relevant to explaining acceptance. VAC only has a positive significant impact on the median (cr = 0.229, *p* = 0.0442, 95%CI = [0.162, 0.704]).Two sociodemographic factors, age and gender, seem to be the most relevant variables to explain UTILITY; whereas, age shows a significant positive relation with UTILITY in the median (cr = 0.229, *p* = 0.0442, 95%CI = [0.007, 0.451]) and at quantile τ = 0.75 (cr = 0.19, *p* = 0.0077, 95%CI = [0.051, 0.329]), being female is positively linked with a positive perception for the COVID-19 passport in the median (cr = 0.455, *p* < 0.0001, 95%CI = [0.243, 0.667]) and quantiles τ = 0.25 (cr = 0.427, *p* = 0.0045, 95%CI = [0.134, 0.719]).Cultural status also has a positive impact on UTILITY in the upper quantile (cr = 0.192, *p* = 0.0065, 95%CI = [0.054, 0.329]).

## 4. Discussion

Regarding the first research question, is people’s utility perception toward IP different depending on the use (traveling regulation vs. controlling access to public places and social activities) that is going to be given to that document, we have found that the IP has a slightly better evaluation when it is used to regulate travel than for controlling assembly in public spaces. However, this better perception is only significant in the case of the third item, “IP are inefficient/efficient”. Likewise, we have checked that the way input variables influence UTILITY is different. VAC is the key variable to explain UTILITY towards the IP to rule mobility since it is significant in the OLS regression and in all LQRs. On the other hand, if the purpose is to control access to public places, the positive impact of VAC on UTILITY is only relevant in its central measures (mean and median). Likewise, whereas LQRs show that factors related to individual status toward COVID-19 are the most significant variables to explain UTILITY toward the use of an IP to rule mobility, gender and age showed a greater impact on UTILITY when we inquired about the use of COVID-19 certificates to control infection spread during leisure activities.

The second research question assessed in this paper was “What are the factors (gender, age, educational level, and those related to the circumstances that suppose sterility in the transmission of the disease) relevant to explain the perception about IP?” We have checked that the main variable explaining the sign of the judgments about the IP is vaccination status. We found it relevant in all regressions when the objective was traveling regulation, and in the central measures of UTILITY, if the objective was controlling assembly. 

Having natural immunization towards COVID-19 due to being infected in the past, is not linked positively with UTILITY. In fact, the regression coefficient of NATIM is usually negative, and this sign shows significance in the LQR adjusted to whether the IP is used to rule mobility at the median. Thus, being infected in the past by COVID-19, and thus, causing immunization does not seem to be an incentive for irresponsible behaviors from those tending to suffer the infection, contrary to what is suggested in [45]. Instead, this result is compatible with the rejection of tracking apps by infected people due to the so-called stigmatization effect [33]. Notice, however, that although they are not COVID-19 transmitters after recovering from the infection, this does not mean they have overcome the symptoms of the disease because of the existence of so-called long Covid [46].

We have observed that women are more favorable than men to the use of IPs to control disease transmission in public spaces. This result is consistent with the mainstream reports on the influence of gender on the acceptance of control measures against COVID-19. They usually report more favorable perceptions by females [23,24,25,26,28,30,34]. However, we must also point out that our findings contradict [37,38]. These papers observed that women have a less favorable perception of immunity passport programs.

The LQR quantiles at τ = 0.5 and τ = 0.75 detect a positive impact of the interviewee’s age on his/her UTILITY towards a mandatory use in a leisure context. This finding is compatible with a great deal of reports outlining that an older age implies greater knowledge and acceptance of measures against COVID-19 [22,23,25,26,30,31,35]. Likewise, interviewees’ cultural status has a positive significant impact on UTILITY at the τ = 0.75 level in both uses of the IP. The literature often finds a significant positive link between the cultural level and the acceptance and application of control measures against COVID-19 [21,22,23,24,26,29,30,31]. We must also recognize that in what refers to the IPs, [37,38] did not find a significant relationship between cultural status and the acceptance of COVID-19 certificates.

## 5. Conclusions and Limitations

To implement the IP effectively, it is advisable to garner a general agreement among the population on its convenience. Therefore, we feel that the results in this study have important implications for public health decision-makers, that they may take into account when conducting work on the IPs. If a mandatory IP is used to control SARS-CoV-2 spread, due to mobility, being vaccinated, and regularly getting tested for the infection, this generates a positive UTILITY toward an IP. Therefore, the implementation of an IP for this purpose, requires that public health authorities ensure easy vaccination and testing for citizens. Therefore, using the IP to regulate travel does not seem to be acceptable to populations from regions without a great coverage of vaccines and tests. Likewise, we have checked that allowing people to obtain an IP due to a past infection does not stimulate irresponsible behaviors. Therefore, we have not found any problem in considering natural immunity as a reason to obtain an IP.

The fact that age and cultural status positively impact UTILITY regarding whether an IP is used to regulate access to public spaces, suggests that a greater knowledge of COVID-19 and its prophylactic measures positively impacts the perception that an IP is useful to prevent COVID-19 from spreading. Therefore, effective implementation of an IP to regulate leisure activities requires convincing the population of the risks of COVID-19 and effective communication strategies of SARS-CoV-2 prophylactic measures.

Quantile regression allows estimating conditional quantiles of a response variable and therefore provides a more complete view of the causal relations between sociodemographic and immunity factors with the utility perception of an IP program. In regression models, whose error term has a heterogeneous variance, it is common that some variables are significant in some quantiles but not in others [44]. This fact is relevant information, which allows a deeper knowledge of how explanatory variables impact judgements about an IP. For example, a variable that is significant, which explains UTILITY at a level of τ = 0.25 (τ = 0.75) but not at τ = 0.5, or in OLS regression, therefore, could be considered as a relevant factor to explain the extreme responses that underrate (overrate) the perception of the IP’s utility in a given sociodemographic and immunity situation but not to explain average responses under that situation.

The cross-sectional survey in this study is not so great (400 observations). There were complaints in April 2021. In that month, IP policies were still not implemented. Drawing a more complete picture of utility perceptions about IP usefulness requires longitudinal surveys covering posterior phases of the pandemic. Likewise, this study is limited to Spain, where there was a high vaccine coverage by the end of 2021 [16]; Spain is one of the countries with the greatest sensitivity toward the urgency posed by COVID-19 [25]. The results may show new nuances in countries and regions with different contextual situations in regard to COVID-19. 

Note, however, that the way in which we have defined some explanatory variables can suppose limitations. With regard to age, we discriminated those age groups with a risk of severe illness due to COVID-19 (at least 45 years) from those whose that risk is low, less than 45 years, [40]. However, this design cannot reflect a possible resistance to the IP that is located in the medium age groups. This fact has been reported for other prophylactic measures, for example, getting vaccinated [47]. Moreover, note that, at the time in which the survey was done, COVID-19 vaccines were not fully available for all the population. Therefore, the variable VAC cannot discriminate between people that were not vaccinated but who were willing to be vaccinated and those that will reject being vaccinated in any case. Therefore, further research to link attitudes toward being vaccinated and perception of the IP constraints must be done.

Likewise, we have to outline that the rate of surveyed people reporting university education (59.7%) is above the proportion of the Spanish population aged between 25–54 years, with a high educational level of 43.3% [48]. However, we do not feel that this circumstance could induce significant biases in our conclusions. Note, however, that RQ1 is answered by comparing the mean and median evaluations and results in OLS and LQR at τ = 0.5, suggesting that cultural status does not have a significant impact on utility perception on mean and median responses. Likewise, in the regression analysis run to assess RQ2, the number of observations reporting university and non-university academic degrees (239 vs 161 observations) was large enough to attain reliable results about the significance of such variable. 

## Figures and Tables

**Table 1 behavsci-12-00140-t001:** Sociodemographic characteristics and immunity situations of respondents.

Variable	Observation
GENDER	50% men and 50% women
AGE	18–24 years = 25% 25–44 years = 25% 45–64 years = 25% plus 64 years = 25% median = 44.5 years mean = 45.38 years standard deviation = 19.87 years
CULTURAL STATUS	59.5% of people have a university degree
VACCINATION STATUS	21.25% of people were vaccinated
NATURAL IMMUNITY	22.5% of people are naturally immunized
NUMBER OF PCRs	67.75% of people reported getting tested at least once

**Table 2 behavsci-12-00140-t002:** Descriptive statistics of items about utility perception and results of Student’s *t*-test on mean differences and Wilcoxon sign-ranked test.

	Traveling			Leisure			Tests on Differences	
Question	Mean	Std. Dev.	Median	Mean	Std. Dev.	Median	*t*-Test	WSRT
Question 1: I feel that implementing IP as a prophylactic measure is a bad/good idea	5.97	3.43	7	5.78	3.63	7	1.442 (0.399)	0.957 (0.339)
Question 2: I feel that implementing IP as a prophylactic measure is a foolish/wise idea	5.95	3.40	7	5.74	3.54	6	1.674 (0.095)	1.265 (0.206)
Question 3: I feel that implementing IP as a prophylactic measure is an Ineffective/effective idea	6.17	3.23	7	5.88	3.40	7	2.606 (0.009)	2.338 (0.019)
Question 4: I feel that implementing IP as a prophylactic measure is negative/positive idea	6.20	3.21	7	6.05	3.41	7	1.419 (0.339)	0.807 (0.419)

Note: In parentheses are *p*-values.

**Table 3 behavsci-12-00140-t003:** Testing the internal consistency of the scale used to measure utility perception.

	Traveling Purpose			Leisure Purpose		
Question	Loading	CA	AVE	Loading	CA	AVE
		0.953	0.834		0.969	0.888
Question 1: I feel that implementing IP as a prophylactic measure is a bad/good idea	0.904			0.929		
Question 2: I feel that implementing IP as a prophylactic measure is a foolish/wise idea	0.907			0.939		
Question 3: I feel that implementing IP as a prophylactic measure is an ineffective/effective idea	0.872			0.928		
Question 4: I feel that implementing IP as a prophylactic measure is negative/positive idea	0.821			0.868		

**Table 4 behavsci-12-00140-t004:** Results of adjusting the regression model with OLS.

	Traveling Purpose			Leisure Purpose		
Variable	Regression Coefficient	95%CI	*p*-Value	Regression Coefficient	95%CI	*p*-Value
Intercept	−0.098	[−0.307, 0.111]	0.357	−0.22	[−0.430 −0.010]	0.041
GENDER	0.097	[−0.097, 0.291]	0.328	0.21	[0.013, 0.407]	0.038
AGE	−0.090	[−0.293, 0.113]	0.386	0.09	[−0.115, 0.295]	0.391
CS	0.059	[−0.140, 0.258]	0.565	0.01	[−0.270, 0.290]	0.942
VAC	0.415	[0.165, 0.665]	0.001	0.27	[0.023, 0.517]	0.033
NATIM	−0.212	[−0.451, 0.027]	0.082	−0.07	[−0.307, 0.167]	0.563
NPCR	0.074	[−0.156, 0.304]	0.531	0.09	[−0.133, 0.313]	0.431
	R^2^ = 0.0376			R^2^ = 0.0299		
	Snedecor’s F = 2.56 (0.0229)			Snedecor’s F = 2.02 (0.062)		
	White’s LM = 35.826 (0.0229)			White’s LM = 31.529 (0.0643)		
	Normality (χ^2^) = 89.972 (<0.001)			Normality (χ^2^) = 76.597 (<0.001)		

**Table 5 behavsci-12-00140-t005:** Linear quantile regressions (IP is used to regulate travel).

Level	τ = 0.25		τ = 0.5		τ = 0.75	
Variable	Regression Coefficient	*p*-Value	Regression Coefficient	*p*-Value	Regression Coefficient	*p*-Value
Intercept	−0.829	<0.0001	0.060	0.6712	0.701	<0.0001
GENDER	0.238	0.1581	0.080	0.545	0.002	0.9806
AGE	−0.239	0.1751	−0.156	0.2619	0.038	0.6363
CS	−0.077	0.6599	0.234	0.089	0.201	0.0106
VAC	0.652	0.0026	0.482	0.0047	0.201	0.0385
NATIM	−0.321	0.1193	−0.327	0.0446	−0.161	0.0833
NPCR	0.484	0.0162	0.078	0.6226	−0.119	0.1895
	95%CI		95%CI		95%CI	
Intercept	[−1.182, −0.476]		[−0.218, 0.338]		[0.542, 0.860]	
GENDER	[−0.092, 0.567]		[−0.179, 0.340]		[−0.147, 0.150]	
AGE	[−0.585, 0.106]		[−0.428, 0.116]		[−0.118, 0.193]	
CS	[−0.418, 0.264]		[−0.035, 0.502]		[0.048, 0.355]	
VAC	[0.230, 1.074]		[0.150, 0.815]		[0.011, 0.391]	
NATIM	[−0.725, 0.082]		[−0.645, −0.009]		[−0.343, 0.021]	
NPCR	[0.091, 0.877]		[−0.232, 0.387]		[−0.296, 0.058]	
Pseudo-R^2^	0.08241		0.03266		0.03762	

**Table 6 behavsci-12-00140-t006:** Linear quantile regressions (IP is used to control access to public activities).

Level	τ = 0.25		τ = 0.5		τ = 0.75	
Variable	Regression Coefficient	*p*-Value	Regression Coefficient	*p*-Value	Regression Coefficient	*p*-Value
Intercept	−0.971	<0.0001	−0.126	0.2773	0.717	<0.0001
GENDER	0.427	0.0045	0.455	<0.0001	0.031	0.6511
AGE	0.023	0.8853	0.229	0.0442	0.190	0.0077
CS	−0.184	0.2331	−0.057	0.6096	0.192	0.0065
VAC	0.286	0.1351	0.433	0.0019	0.040	0.6438
NATIM	−0.178	0.3293	−0.220	0.0975	0.067	0.4199
NPCR	0.297	0.0957	0.082	0.5259	0.042	0.6044
	95%CI		95%CI		95%CI	
Intercept	[−1.285, −0.658]		[−0.353, 0.101]		[0.575, 0.859]	
GENDER	[0.134, 0.719]		[0.243, 0.667]		[−0.102, 0.163]	
AGE	[−0.284, 0.329]		[0.007, 0.451]		[0.051, 0.329]	
CS	[−0.487, 0.118]		[−0.276, 0.162]		[0.054, 0.329]	
VAC	[−0.088, 0.660]		[0.162, 0.704]		[−0.130, 0.210]	
NATIM	[−0.537, 0.180]		[−0.479, 0.040]		[−0.096, 0.229]	
NPCR	[−0.052, 0.646]		[−0.171, 0.335]		[−0.116, 0.200]	
Pseudo-R^2^	0.05550		0.03231		0.04617	

## Data Availability

The data is available by correspondence with the author.

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
