# Peer review of "Citizens’ Perception of COVID-19 Passport Usefulness: A Cross Sectional Study"

_behavsci, 2022, doi:10.3390/bs12050140_

Round 1

Reviewer 1 Report

General

Authors explore in their manuscript ‘Sociodemographic Factors, Immunity Situations And Citizens’

Perception Of Covid-19 Passport Usefulness: A Cross Sectional Study’ the relationships which exist between several factors and the Covid-19 passport usefulness. Such topics are understudied, however some work should be done before this manuscript can be published.

Title

I think the title is a bit too lengthy; perhaps skip the words ‘Immunity Situations’ or make of it: ‘Citizens’

Perception Of Covid-19 Passport Usefulness measured with some selected factors: A Cross Sectional Study’

Abstract

Background

Please change <of its use:> to ‘of its use.

Methods

Were it exactly <400 residents>?

Results

-

Conclusion

-

Key words

-

Introduction

Please change <COVID-19 is possible> into ‘COVID-19 was possible’

Please change <to limit mobility and assembly have been the only> into ‘to limit mobility and assembly were the only’

Please change <Thus, since the rise of the SARS-COV-2> into ‘Thus, since the rise of the SARS-CoV-2’

Please change <a negative test result [8] Different> into ‘a negative test result [8]. Different’

Please explain what means <IATA>

Methods

Sample

400 – explain the 400; was it ‘by accident’ or did you aim at this number?

Please explain why so many <59.5% people have a university degree> people have a university degree?? Or is this a limitation?

Is the high number of people having a university degree related to the rather low number of people being vaccinated?

Measures

Please change <(1) a bad/good idea (2) a foolish/wise idea (3) an innefective/effective measure (4) a negative/positive measure> into ‘(1) a bad/good idea; (2) a foolish/wise idea; (3) an inneffective/effective measure; (4) a negative/positive measure’

Please explain <standard deviation=19.87 years.> (Table 1) the possibility of a that low SD (taken into account that 59.5% has a university degree)

Statistical analyses

RQ2: Please rewrite this section: As third step, we … . (fourth?) Finally, we … . The readership easier grabs what you did and in which order the Results will be shown.

Results

-

Discussion

-

Tables, Figures

-

References

Author Response

Perception Of Covid-19 Passport Usefulness: A Cross Sectional Study’ the relationships which exist between several factors and the Covid-19 passport usefulness. Such topics are understudied, however some work should be done before this manuscript can be published.

Response: We thank your valuable comments that have allowed improving notably the paper.

Title

I think the title is a bit too lengthy; perhaps skip the words ‘Immunity Situations’ or make of it: ‘Citizens’

Perception Of Covid-19 Passport Usefulness measured with some selected factors: A Cross Sectional Study’

Response: We have simplified the tittle to “Citizens’ Perception of Covid-19 Passport Usefulness: A Cross Sectional Study”

Abstract

Background

Please change <of its use:> to ‘of its use.’

Response: We have done so.

Methods

Were it exactly <400 residents>?

Response: Yes. See in the new version lines 136 and ff that are highlighted.

Introduction

Please change <COVID-19 is possible> into ‘COVID-19 was possible’

Response: Done (see lines 30 and ff)

Please change <to limit mobility and assembly have been the only> into ‘to limit mobility and assembly were the only’

Response: Done (see lines 30 and ff)

Please change <Thus, since the rise of the SARS-COV-2> into ‘Thus, since the rise of the SARS-CoV-2’

Response: Done. We have done so throughout the document.

Please change <a negative test result [8] Different> into ‘a negative test result [8]. Different’

Response: Done. We have done so.

Please explain what means <IATA>

Response: Done. We have done so (line 49)

Methods

Sample

400 – explain the 400; was it ‘by accident’ or did you aim at this number?

Response: We have explained it. See in the new version lines 136 and ff that are highlighted.

Please explain why so many <59.5% people have a university degree> people have a university degree?? Or is this a limitation?

Response: We have addressed this query in limitations (las paragraph in the paper, line 401 and ff, blue highlighted)

Is the high number of people having a university degree related to the rather low number of people being vaccinated?

Response: We have addressed this query lines 147 and ff (blue highlighted).

Measures

Please change <(1) a bad/good idea (2) a foolish/wise idea (3) an innefective/effective measure (4) a negative/positive measure> into ‘(1) a bad/good idea; (2) a foolish/wise idea; (3) an inneffective/effective measure; (4) a negative/positive measure’

Response: We have done so (lines 160 and ff)

Please explain <standard deviation=19.87 years.> (Table 1) the possibility of a that low SD (taken into account that 59.5% has a university degree)

Response: We have addressed this question (see lines 139 and ff blue highlighted)

Statistical analyses

RQ2: Please rewrite this section: As third step, we … . (fourth?) Finally, we … . The readership easier grabs what you did and in which order the Results will be shown.

Response: We have done so (see lines 207-224 blue highlighted)

Reviewer 2 Report

The article deals with an important public health problem. The paper made statistical analysis of 400 individual answers. The data belongs to April of 2021 when vaccination was not fully available.

It is important to mention the main limitations of this study. For instance, how the explanatory variables are designed. In addition, the fact that some people were willing to vaccinate but not vaccine availability. These important factors can affect the statistics results. Since vaccine availability could be an issue, this can affect the results. What happens if that VAC variable is removed? How the other variables vary in this case?

The questions seem very similar. Explain in the manuscript why they are so similar and what the aim of that is. Maybe in Spanish the words can be interpreted on a different way?

In the conclusions, it is important to mention the importance and differences of the results related to tau. For instance explain the meaning that a variable is significant for tau=0.75, but no for other taus.

Explain in the manuscript why the R2 pseudo was used.

Author Response

The article deals with an important public health problem. The paper made statistical analysis of 400 individual answers. The data belongs to April of 2021 when vaccination was not fully available.

Response: Thanks a lot for your helpful comments that have allow improving notably the paper.

It is important to mention the main limitations of this study. For instance, how the explanatory variables are designed. In addition, the fact that some people were willing to vaccinate but not vaccine availability. These important factors can affect the statistics results. Since vaccine availability could be an issue, this can affect the results. What happens if that VAC variable is removed? How the other variables vary in this case?

Response: We have addressed these questions in last three paragraphs of our paper (lines 383-410) where we expose several limitations.

The questions seem very similar. Explain in the manuscript why they are so similar and what the aim of that is. Maybe in Spanish the words can be interpreted on a different way?

Response: Questions reflect several nuances of the construct “perceived usefulness”. We have addressed this question in lines 154-167 and 243-245 (yellow highlighted).

In the conclusions, it is important to mention the importance and differences of the results related to tau. For instance explain the meaning that a variable is significant for tau=0.75, but no for other taus.

Response: We have explained it in conclusions, lines 372-382 (green highlighted).

Explain in the manuscript why the R2 pseudo was used.

Response: We have addressed it in lines 267-271 (yellow highlighted)

Round 2

Reviewer 1 Report

General

Authors explore in their manuscript ‘Citizens’ Perception of Covid-19 Passport Usefulness: A Cross Sectional Study’ the relationships which exist between several factors and the Covid-19 passport usefulness. Such topics are understudied, this manuscript can (according to me) be published.

Methods

Sample

Please explain <health workers o teachers>. It is a typo, but I don't know what you mean.

Measures

Please explain <standard deviation=19.87 years.> (Table 1) the possibility of a that low SD (taken into account that 59.5% has a university degree). It is in all EU countries lower.

Statistical analyses

Delete one of both <we fit we fit>

Author Response

Authors explore in their manuscript ‘Citizens’ Perception of Covid-19 Passport Usefulness: A Cross Sectional Study’ the relationships which exist between several factors and the Covid-19 passport usefulness. Such topics are understudied, this manuscript can (according to me) be published.

Response: Thanks a lot for your suggestions that improved the paper.

Please explain <health workers o teachers>. It is a typo, but I don't know what you mean.

Response: This was a typo. We have deleted “o teachers”.

Please explain <standard deviation=19.87 years.> (Table 1) the possibility of a that low SD (taken into account that 59.5% has a university degree). It is in all EU countries lower.

Response: We expanded this issue in lines 141-152 (see yellow highlighted)

Delete one of both <we fit we fit>

Response: We have solved the typo

Reviewer 2 Report

The authors improved the paper taking into account the reviewers' comments.

Author Response

The authors improved the paper taking into account the reviewers' comments.

Response: Thanks a lot for your suggestions that improved the paper.